# Fast and Accurate Surrogate Virus Neutralization Test Based on Antibody-Mediated Blocking of the Interaction of ACE2 and SARS-CoV-2 Spike Protein RBD

**DOI:** 10.3390/diagnostics12020393

**Published:** 2022-02-03

**Authors:** Denis E. Kolesov, Maria V. Sinegubova, Lutsia K. Dayanova, Inna V. Dolzhikova, Ivan I. Vorobiev, Nadezhda A. Orlova

**Affiliations:** 1Laboratory of Mammalian Cell Bioengineering, Skryabin Institute of Bioengineering, Research Center of Biotechnology of the Russian Academy of Sciences, 117312 Moscow, Russia; 52ru111@mail.ru (D.E.K.); mvsineg@gmail.com (M.V.S.); l.daianowa@yandex.ru (L.K.D.); ptichman@gmail.com (I.I.V.); 2Laboratory of Glycoproteins Biotechnology, Institute of Bioorganic Chemistry of the Russian Academy of Sciences, 117997 Moscow, Russia; 3Gamaleya National Research Center of Epidemiology and Microbiology, Ministry of Healthcare of the Russian Federation, 123098 Moscow, Russia; i.dolzhikova@gmail.com

**Keywords:** sVNT, SARS-CoV-2, ACE2

## Abstract

The humoral response to the SARS-CoV-2 S protein determines the development of protective immunity against this infection. The standard neutralizing antibodies detection method is a live virus neutralization test. It can be replaced with an ELISA-based surrogate virus neutralization test (sVNT), measuring the ability of serum antibodies to inhibit complex formation between the receptor-binding domain (RBD) of the S protein and the cellular ACE2 receptor. There are conflicting research data on the sVNT methodology and the reliability of its results. We show that the performance of sVNT dramatically improves when the intact RBD from the Wuhan-Hu-1 virus variant is used as the plate coating reagent, and the HRP-conjugated soluble ACE2 is used as the detection reagent. This design omits the pre-incubation step in separate tubes or separate microplate and allows the simple quantification of the results using the linear regression, utilizing only 3–4 test sample dilutions. When this sVNT was performed for 73 convalescent plasma samples, its results showed a very strong correlation with VNT (Spearman’s Rho 0.83). For the RBD, bearing three amino acid substitutions and corresponding to the SARS-CoV-2 beta variant, the inhibitory strength was diminished for 18 out of 20 randomly chosen serum samples, and the magnitude of this decrease was not similar to the change in overall anti-RBD IgG level. The sVNT assay design with the ACE2-HRP is preferable over the assay with the RBD-HRP reagent and is suitable for mass screening of neutralizing antibodies titers.

## 1. Introduction

The importance of serological testing for anti-SARS-CoV-2 humoral immune response in clinical practice, as well as for public health, is demonstrated by the huge variety of test systems currently approved worldwide [1]. However, most of them can quantitatively or qualitatively determine the level of IgG and IgM antibodies toward SARS-CoV-2 antigens without considering the functional features of these antibodies.

Many of the currently authorized anti-SARS-CoV-2 vaccines demonstrated more than 90% efficacy against the COVID-19 [2,3], it was found that the achieved protective immunity strongly correlates with the level of virus-neutralizing antibodies (nAb) [4,5]. Obviously, anti-SARS-CoV-2 antibody and nAb titers decline over time after vaccination [6] or natural infectious events [7], so achieving the desired herd immunity might require further vaccinations in the coming years. Despite the introduction of WHO international standards for anti-SARS-CoV-2 immunoglobulins, it is currently unclear what nAb or anti-RBD antibody levels are sufficient to prevent SARS-CoV-2 infection and reinfection. This issue can be resolved using mass quantitative testing of nAb levels in the general population. Such testing will require a reliable and easy-to-run testing procedure.

It should be noted that although the nAbs are a subset of general antigen-binding antibodies, and in the case of SARS-CoV-2, they are anti-S-protein antibodies (Figure 1A), nAb levels do not correlate well with total Ab titers and cannot be deduced from total Ab titers [8]. For example, in the study of COVID-19 convalescent individuals (n = 149), the correlation coefficient of only 0.64 was obtained for nAb levels and anti-RBD antibodies levels, with the appearance of a large subset of nAb-negative samples, possessing medium to high anti-RBD Ab titers [9]. This limited correlation is based on the variance of humoral immune response rise and decline between individuals [10], despite the generation of natural nAb clones mainly as the anti-RBD antibodies [11].

The most realistic nAb testing method is the plaque-reduction neutralization test (PRNT), performed as the addition of laboratory-adapted live virus to the cultured virus-susceptible cells and subsequent counting of the virus-induced plaques in the cells monolayer. Virus neutralization by antibodies is usually referred to as the last serial dilution of the serum sample, giving the more than 50% reduction in plaque numbers, so the PRNT results are semi-quantitative values. For the SARS-CoV-2 virus, PRNT is usually based on HEK293 cells overexpressing the cellular receptor ACE2 [12], which binds to the viral S protein, and in some cases additionally overexpressing the TMPRSS2 protease [13]. This membrane-bound serine protease cleaves the S-protein and initiates its conversion to a membrane-protruding conformation.

The PRNT is considered the reference VNT method, but it requires a Biosafety Level 3 (BSL-3) laboratory, takes at least two working days, and is very labor-consuming, making it impossible to use this test for any mass screenings [12]. The live virus could be substituted to the less dangerous pseudotyped virus, for example, the replication-competent vesical stomatitis virus with the S-protein from SARS-CoV-2 and the reporter gene of the firefly luciferase, resulting in the pseudotyped virus neutralization test (pVNT) [14]. This substitutive test can be performed in much more common BSL-2 laboratories but still requires a live cell culture, unusual analytical equipment, and 2–3 days to complete, so the routine mass use of pVNT is questionable. Further virtualization of the virus used in the VNT assay might reduce the biosafety requirements for the testing facility but would not change the time of the assay and associated labor and material costs.

The surrogate VNT (sVNT) measures the blocking of the ligand–receptor complex formation upon the addition of the test serum samples. For SARS-CoV-2, the test reagents are RBD or S1 subunits of the S-protein and soluble human ACE2, and the testing procedure is a conventional ELISA, so the sVNT is best suited for the clinical setting and allows rapid plasma sample screening in the laboratories with low biosecurity level.

In most studies published to date, the SARS-CoV-2 sVNT is based on the intact soluble ACE2, immobilized on microplates, and the RBD-HRP conjugate, used as the detection reagent. This test design was first described in [15], is available commercially as the ready-to-use reagent kit, and was recommended, along with many other direct ELISA kits, for the selection of convalescent plasma samples for COVID-19 therapy.

However, several studies report that sVNT with the plate-immobilized ACE2 does not show a consistent correlation with real virus neutralization tests, and thereby cannot be a reliable alternative to the VNT and pVNT [16,17].

There are two general designs of the SARS-CoV-2 sVNT, including the intact soluble ACE2, immobilized on the microplate, and the RBD-HRP conjugate as the detection reagent (ACE2:RBD-HRP) or intact RBD as the coating reagent and the ACE2-HRP as the detection reagent (RBD:ACE2-HRP) (Figure 1C). The second design is mentioned in a few publications and was implemented as a three-layer sandwich ELISA with the ACE2-biotin reagent and subsequent avidin-HRP conjugate incubation [18].

Although both sVNT designs should give the same or similar results as they measure inhibition of the same ligand–receptor complex formation, they might be not equivalent for a number of reasons—e.g., due to different steric availability of the immobilized reagent, formation of signal-enhancing immune complexes in solution or on the microplate surface, reagent instability, lack of equilibrium, or other technical issues.

We decided to directly compare both sVNT designs using the same in-house prepared reagents and then determine the limitations and usefulness of sVNT, comparing the results of an optimal sVNT design with those of live VNT. The ultimate goal of this work is to develop a simple and reproducible technique for sVNT that will retain a good correlation with VNT and allow a wide screening of donated convalescent plasma units for the emergency therapy of COVID-19 as well as inexpensive testing of the general population for nAb levels. 

## 2. Materials and Methods

### 2.1. Molecular Cloning

pTF-ACE2s construction. cDNA corresponding to the human ACE2 ORF was a generous gift of prof Peter Sergiev, MSU, Moscow, Russia. The extracellular domain of the ACE2 ORF (Figure 1B) was amplified using Tersus polymerase mix and adaptor primers AD-ACE-AbsF and AD-ACE-NheR (Table A1). The PCR product was cut with AbsI and AsuNHI and cloned into the pTF vector, which was digested with the same enzymes. The pTF vector is a derivative of the p1.1-Tr2-eGFP plasmid (GenBank: MW187857). The pTF polylinker was constructed from the pair of annealed 5’phosphorylated oligos AS-Flag-AbsF and AS-Flag-AbsR (Appendix A, Table A1); the reaction conditions are described in detail in [19]. The pTF vector and pTF-ACE2s plasmid are available from Addgene—plasmid nos. 162784 and 162786. The oligos used for the expression plasmid sequencing, SQ-5CH6-F, SQ-ACE875-R, SQ-ACE592-F, SQ-ACE1170-F, SQ-ACE1512-F, are listed in Table A1.

pTM-RBDv2_NKY (beta B.1.351) construction. The N-terminal part of the RBD open reading frame was removed from the expression plasmid pTM-RBDv2 encoding Wuhan-Hu-1 S-protein RBD [19] by restriction with AbsI and SphI enzymes. This region was replaced with a synthetic AbsI-SphI DNA fragment encoding the same polypeptide, with one substitution in the signal peptide S[−1]−>R[−1] and three mutations in the mature RBD polypeptide—K417N, E484K, N501Y. Thus, the expressed protein corresponded to the beta variant of SARS-CoV-2 (B.1.351) (Figure 1B).

Purified plasmids were concentrated by ethanol precipitation in sterile conditions and used for cell transfection.

The synthetic DNA was obtained from the Cloning Facility, Moscow, Russia. The restriction enzymes were from Sibenzyme, Novosibirsk, Russia. All the other reagents used for cloning, including the synthetic oligos, PCR-mix, Plasmid Miniprep Purification kit, PCR Clean-Up System, and Plasmid Midiprep kit were from Evrogen, Moscow, Russia. 

### 2.2. Producer Cell Lines Generation and Cell Culture

A DHFR-negative Chinese hamster ovary (CHO) DG-44 cell line (Thermo Fischer Scientific, Waltham, MA, USA) was used as a host cell line [20]. The cells were cultivated as a suspension in 125 mL Erlenmeyer flasks (Corning, NY, USA) containing 30 mL of ProCHO5 culture medium (Lonza, Switzerland), 4 mM glutamine, 4 mM alanyl-glutamine, and hypoxanthine-thymidine (HT) supplement (all PanEco, Moscow, Russia). The pTF-ACE2s plasmid was transfected into the CHO DG44 cells using nucleofection with Neon apparatus and the Neon transfection kit (both Thermo Fisher Scientific, Waltham, MA, USA) according to the kit manufacturer instructions. Stably transfected cell population (cell pool) was obtained by selection medium without HT and in the presence of 200 nM methotrexate (MTX). The initial stably transfected cell pool was subjected to two consecutive rounds of the target gene amplification in the presence of 2 μM and 8 μM MTX. Total selection and amplification time was about 3 months. The cells were passaged every 3 to 5 days at each stage until the cell viability was restored to 80%. The final cell pool was used for single-cell cloning by serial dilution method, performed as described previously [19], resulting clonal cell line with highest specific productivity was used for the ACE2 production. Beta-RBD-producing cell line was obtained as described above, polyclonal cell population, obtained after two rounds of gene amplification was utilized for beta-RBD protein production.

### 2.3. Preparative Cell Cultures

Preparative cultivation was performed in the absence of selection pressure in a fed-batch mode. The medium was Lonza ProCHO5 supplemented with 4 mM glutamine, 4 mM alanyl-glutamine. Glucose level was measured daily since day 5 with the Accutrend Plus System (Roche, Switzerland) and kept in a 20–50 mM range. Seeding cell culture was grown first in 125 mL Erlenmeyer flasks in a volume of 30 mL for 3–4 days to achieve cell density around 1 × 10^6^ cells/mL, then split at a ratio 1:3 in 250 mL Erlenmeyer flasks in a volume of 60 mL and cultivated for additional 7–10 days. The cells were harvested when the viability measured using the trypan blue exclusion fell below 50–60%.

### 2.4. Western Blotting

Conditioned culture medium from the initial stably transfected cell population, secreting the ACE2, was concentrated using ultrafiltration 100× and buffer-exchanged to PBS using a 10 kDa MWCO Vivaspin 500 concentrator (Sartorius, Göttingen, Germany) according to the manufacturer instructions. SDS-PAGE, protein transfer, and Western blotting were performed as described in [19] with minor changes—separation gel contained 10% acrylamide and anti-FLAG mAb—HRP conjugate (A8592, Sigma, Burlington, MA, USA, dilution 1:10000) was used for blotting.

### 2.5. Proteins Purification

ACE2 was purified from the conditioned culture medium using the immobilized metal affinity chromatography, as described in [19] with some changes. Clarified conditioned medium was supplemented with 300 mM NaCl, 50 mM Tris pH 8.0, and directly applied to the column with Ni-chelate Sepharose (Cytiva, Marlborough, MA, USA). Column wash solution was 10 mM imidazole, 300 mM NaCl, 50 mM Tris-HCl, pH 8.0. The target protein was eluted with the 50 mM imidazole, 300 mM NaCl, 50 mM Tris-HCl, pH 8.0 solution. The semi-purified ACE2 was additionally purified using the anion exchange chromatography, 1 mL HiTrap Q Fast Flow column (Cytiva, Marlborough, MA, USA). Eluate of the Ni-chelate Sepharose column was diluted with 1 volume of deionized water and applied to the Q resin at 1 mL/min. The column was washed with the 20 mM sodium phosphate pH 7.4 solution, and the target protein was eluted with the NaCl concentration gradient, from 0 to 1 M in 20 column volumes. Purified ACE2 was concentrated and buffer-exchanged to the 20 mM sodium phosphate, pH 7.4; 250 mM NaCl using the 10 kDa MWCO PES Vivaspin 500 concentrators (Sartorius, Göttingen, Germany) and stored frozen in aliquots. RBD-NKY protein was purified from a conditioned medium exactly as described in [19]. SARS-CoV-2 nucleoprotein (NP), tagged with 10× His was purified in-house from the *E. coli* cytoplasm as described in [21], absence of RNA was controlled by the Qubit RNA assay (ThermoFischer Scientific, Waltham, MA, USA).

### 2.6. Recombinant Proteins Analysis

Preparations of the RBD, ACE2, beta-RBD, and NP were routinely tested for purity using the SDS-PAGE with colloidal Coomassie staining and size-exclusion chromatography, as described in [19]. The binding of ACE2 and RBD was studied with the surface plasmon resonance, utilizing the Proteon XPR36 instrument (Bio-Rad Laboratories, Hercules, CA, USA) and the GLM sensor chip (Bio-Rad, Hercules, CA, USA). Both ACE2 and RBD were immobilized on the chip’s channels according to the chip manufacturer’s instructions, at a concentration of 25 μg/mL, applied for 10 min at a flow rate of 30 μL/min pH 4.5 for ACE2, and pH 5.5 for RBD. The sensograms were obtained for the 3.1–100 nM analyte concentrations range for RBD and 0.82–200 nM for ACE2. The dissociation constant was calculated utilizing the instrument’s ProteOn Manager software (Bio-Rad, Hercules, CA, USA), a heterogeneous analyte model.

The amidolytic activity of purified ACE2 was measured with the FRET peptide substrate MCA-YVADAPK-(DNP) (M-2195, Bachem, Germany), used at a final concentration of 15 μM. The hydrolysis reaction was set up in the 100 mM Tris-HCl pH 7.5, 0.1% Tween 20, 1 μM ZnSO4, 15 μM substrate solution. ACE2 was added at a final concentration of 5 μg/mL. The fluorescence intensity increase was monitored in kinetics mode for 10 min on a Fluorat-02-Panorama spectrofluorometer (Lumex, St. Petersburg, Russia) at an excitation wavelength of 328 nM and an emission wavelength of 392 nM. The reaction mixture was incubated for 2 h at room temperature outside the spectrofluorometer to achieve full substrate cleavage, diluted 10× with deionized water and used to prepare the MCA concentration—fluorescence intensity calibration curve. One unit of the ACE2 amidolytic enzyme activity was defined as the quantity of substrate, in picomoles, hydrolyzed with the ACE2 in one minute.

### 2.7. HRP Conjugation

ACE2 and RBD were conjugated to the horseradish peroxidase using HRP Conjugation Kit—Lightning-Link (Ab102890, Abcam, Cambridge, UK) as per the manufacturer’s instructions. Conjugates were diluted 2× with the glycerol and stored in a liquid form protected from light at −18 °C.

### 2.8. Sample Collection and VNT Assay

The study was conducted according to the guidelines of the Declaration of Helsinki and approved by the National Research Center for Epidemiology and Microbiology Ethics Committee Protocol No. 17 of 3 December 2021. According to the Ethics Committee protocol, all serum samples used in this study were residuals from the routine laboratory testing, and its further testing does not require informed consent.

Serum samples were collected and the analysis of the neutralizing activity of the convalescent blood plasma was carried out in the microneutralization test described earlier [22]. Live hCoV-19/Russia/Moscow_PMVL-1/2020 virus variant was employed; test was performed as the microneutralization assay with the Vero E6 cells. In total, 73 samples were used for current study. This virus variant contains one amino acid substitution in the S-protein RBD relative to the Wuhan isolate—614D > G.

### 2.9. ELISA

ELISA 96-well plates (Corning, NY, USA) were pre-coated overnight at +4 °C with 100 μL/well of RBD in PBS solution (200 ng/well), or ACE2 in PBS solution (300 ng/well), or ACE2 in 100 mM sodium carbonate-bicarbonate buffer (pH 9.6) (100 μL/well). SARS-CoV-2 NP was used for plate coating at 100 ng/well in PBS. Plates were washed with PBS—0.02% Tween (PBST) thrice, blocked with 250 μL/well of 3% BSA-PBS, washed with PBST and used immediately or sealed with the adhesive film and stored at –18 °C.

The analyzed sera were diluted with the 1% BSA-PBS as 1:2000 (RBD antigen) or as 1:5000 (NP antigen) and incubated in plate wells for 1 h at 37 °C. Wells were washed three times with PBST, secondary anti-human IgG antibody-HRP conjugate (Xema Co., Ltd., Moscow, Russia, cat. T271X@1702) was used at the 1:20000 dilution, the incubation time was 1 h at +37 °C.

Positivity indices (OD/CO) were calculated as the ratio of the mean optical density for the test sample and the mean optical density in negative wells plus three times the standard deviation, according to the [23].

### 2.10. Surrogate Virus Neutralization Test

Test sera were diluted with the 1% BSA-PBS, applied as serial dilutions in the 1:20–1:160 range to the RBD-coated plates (RBD:ACE2-HRP test design) and incubated in plate wells for 30 min at 37 °C, if not stated otherwise in the Results section. ACE2-HRP conjugate was used at 0.7 nM concentration (corresponds to 62.5 ng/mL of the ACE2) in the case of RBD antigen and 2.9 nM (250 ng/mL) in the case of beta-RBD antigen, if not stated otherwise in the Results section. In the case of the ACE2:RBD-HRP test design, test serum samples were diluted as 1:20–1:160, pre-incubated with RBD-HRP in polypropylene test tubes (RBD-HRP concentration 14.2 nM, corresponds to 390 ng/mL of the RBD) for 30 min, transferred to ACE2-coated ELISA plates and incubated further for 30 min.

Wells were washed five times with the PBST. The color was developed for exactly 10 min at room temperature (+25 ± 2 °C), utilizing the ready-to-use TMB solution (Xema Co., Ltd., Moscow, Russia), 100 μL/well. The reaction was stopped with 100 μL of 5% orthophosphoric acid. Optical density at 450 nM was measured using the Multiskan EX plate reader (Thermo Fischer Scientific, Waltham, MA, USA).

### 2.11. Statistical Analysis

All assays were carried out in duplicate, with the relevant positive and negative controls. Statistical analysis was performed using the RStudio software (RStudio PBC, Boston, MA, USA). Correlation and linear regression analyses were performed in RStudio using Spearman correlation coefficients. In the case of Spearman analysis r values of 0–0.19 were regarded as very weak, 0.2–0.39 as weak, 0.40–0.59 as moderate, 0.6–0.79 as strong, and 0.8–1 as very strong correlation. Code for the correlation analysis performed is deposited in GitHub [24].

## 3. Results

### 3.1. Production of Soluble ACE2

The extracellular part of the human ACE2 with C-terminal FLAG and 10× His tags (Figure 1B) was obtained in CHO cells, using the derivative the p1.1 plasmid described in [25]. Polyclonal cell population, created by two subsequent rounds of the MTX-driven genome amplification, secreted ACE2 up to 25 mg/L for 3 days batch culture and the derived most productive clonal cell line possessed the productivity of up to 100 mg/L of the ACE2 for the 7 days long glucose-fed culture. Soluble ACE2 was found to be sufficiently resistant to proteolytic degradation during cell cultivation, according to the Western blot data (Figure 2A). The target protein was purified from the conditioned medium using the immobilized metal affinity chromatography and anion exchange chromatography to the 91% purity as determined by SDS-PAGE (Figure 2B), and less than 1% of multimers as determined using the size exclusion chromatography. (Figure 2C). Amidolytic activity of the ACE2, measured with the MCA-YVADAPK-(Dnp)-OH substrate was 176,000 pmol substrate/ (mg × min), comparable to the previously reported values for recombinant human ACE2 (around 200,000 pmol substrate/(mg × min) in [26] at the same assay conditions). Purified soluble ACE2 was also tested by the surface plasmon resonance (SPR) for SARS-CoV-2 RBD binding and dissociation. Dissociation constant was determined as 34 nM (Figure 2D), also comparable to previously published values, for example, 74 nM in [27]. It is interesting to note that the SPR sensograms demonstrated a reasonable association and dissociation rate of the ACE2-RBD complex only if ACE2 was covalently immobilized on the chip (Figure 2D); in the case of RBD immobilization and utilization of ACE2 solution as the analyte, association and dissociation rates were too slow for any adequate measurements (Figure 2E), resulting in the erroneous calculation of the dissociation constant as 1 nM due to practical absence of analyte desorption in 10 min of monitoring. Similar slow association–dissociation kinetics was also detected for the ELISA format assay, which is discussed in detail below.

### 3.2. ACE2-RBD Interaction in Microplates

We have tested in parallel both ways to assemble the ACE2:RBD complex in the ELISA format—ACE2:RBD-HRP and RBD:ACE2-HRP assay designs (Figure 1C). It was found that RBD-HRP interacts with the immobilized ACE2 fast, and practical equilibrium was achieved in 15 min, but the ACE2-HRP binds to the immobilized RBD very slowly; therefore, inside one hour of incubation, the resulting optical density (OD) in microplate wells increased linearly over time (Figure 3C). The concentrations of RBD-HRP and ACE2-HRP, suitable for OD generation of approximately 2 AU after 10 min of incubation with HRP substrate solution, were 0.7 nM and 14 nM, respectively, both comparable to the dissociation constant. Both RBD-HRP and ACE2-HRP probes demonstrated the linear relationship of OD value and the probe concentration (Figure 3A,B), with a better R^2^ value for the ACE2-HRP probe. Both assay designs are expected to produce adequate quantitative results—the residual OD in experimental wells should be directly proportional to the concentration of free RBD or the quantity of free RBD molecules on the well’s surface.

### 3.3. Surrogate VNT Assay Design Variants

Although the ACE2-HRP detection reagent showed very slow binding kinetics, it retains the main preference—it will not interact with the anti-SARS-CoV-2 antibodies at all, so there is no need for pre-incubation of the serum sample with RBD antigen in a separate tube. There are reports on the emergence of anti-ACE2 IgM antibodies in some severe COVID-19 patients [28] or even the development of non-inhibitory anti-ACE2 autoantibodies in most convalescents [29]. However, in the general population, ACE2 is not known to be a target for the autoimmune response, so the possibility of producing the false-positive sVNT titer due to the presence of ACE2-binding antibodies and absence of anti-RBD antibodies may be considered as low.

In the case of the ACE2:RBD-HRP assay design, test samples should be pre-incubated with RBD-HRP outside the microplate. The mixture of serum and RBD-HRP should be added to ACE2 in the microplate wells only after the immune complex formation process has reached equilibrium, and after addition should be incubated until free RBD-HRP binds to ACE2.

We tested both assay designs with two and three representative convalescent serum samples with high, medium, and very low nAb levels (Figure 3D,E). We found that both designs produce adequate nearly linear inhibition curves, and the RBD:ACE2-HRP design produces more linear curves for medium rates of inhibition. Regular analysis of sVNT data is based on the regression of a multivariable logistic curve obtained for 6–8 serial dilutions of the test sample. An example of such a curve is shown in Figure 3F. These binding inhibition curves are non-linear and unsuitable for simple linear regression analysis.

In the case of immobilized ACE2, we also observed that ACE2-coated microplates are quite unstable during storage at −18 °C; even after two days of storage, the OD readings in wells with the same analytes differed from each other by up to 40% (Figure A1). This instability of the immobilized ACE2 was similar when the antigen immobilization was performed in PBS and sodium carbonate buffered solution. At the same time, microplates with immobilized RBD retained the even analyte binding after six months of storage at −18 °C. ACE2-HRP conjugate, prepared by us using the Lightning Link kit, upon six months of storage in liquid form with 50% glycerol at −18 °C, produced the same RBD binding curve in ELISA (Figure A1).

The assay was further optimized for the serum samples incubation time, preceding the addition of the ACE2-HRP reagent. The observed degree of inhibition slightly increased for up to 30 min of the pre-incubation time. It should also be noted that excluding the stage of pre-incubation of the test samples did not significantly decrease sensitivity (Figure 3G).

### 3.4. Binding Inhibition Curves Analysis

In the RBD:ACE2-HRP assay design, it was possible to plot a linear relationship between the degree of inhibition and the test samples concentration instead of usual S-shaped binding curves. Linear plots are easier to analyze and the generic spreadsheet software may be used for this. (Figure 3E,F). Degree of inhibition (DI) was defined as the ratio of the optical density in the negative control wells (without serum, OD_0_) and the optical density in wells with the diluted test sample (OD_inh_), minus 1.
DI = (OD_0_/OD_inh_ − 1)(1)

By this definition, a DI of one corresponds to a 50% reduction in RBD:ACE2 complex formation.

For many individual convalescent serum samples, it was possible to plot the linear relationship between the serum concentration, in per mille (‰) and the DI (Figure A2), then calculate with the simple linear regression the serum dilution, which should produce the DI = 1.0, i.e., inhibit the RBD:ACE2 complex formation twofold. This calculated inhibitory dilution (ID50) was used for further data analysis and compared with two other variants of the characteristic value—the degree of inhibition, obtained at the 40× or 20× serum dilution (Figure A3).

In addition to serum samples obtained from individual donors, we performed sVNT on composite serum controls from the NIBSC test panel. These samples exhibited mostly the nonlinear dose–response curves, and most of these samples produced very weak inhibitory effects (Figure A4) than those taken at concentrations up to 25‰ (40× dilution).

### 3.5. Correlation of sVNT and Live Virus VNT Values

For 73 convalescent donor serum samples, we determined both the threshold dilutions in the live virus VNT and ID50 dilutions in the sVNT assay (Figure 4A). The sVNT data correlated well with VNT results (Figure 4B). For some samples with good neutralization result in VNT, the sVNT measurement did not show a significant neutralization effect; at the same time, all serum samples with sVNT ID50 > 50× correspond to the effective neutralization in VNT—threshold dilutions 80× or more (Figure 4A,C). The proportion of false-positive samples—i.e., showing inhibitory properties above the proposed 50× cut-off in sVNT and weak inhibitory properties in VNT—is only 1.37% (1 from 73), but the proportion of false-negative samples (low sVNT, high VNT dilution) is large, up to 30%. Most likely, these sera contain mainly nAbs that bind the S protein outside the RBD domain and block the virus entry, regardless of the disruption of the ligand–receptor complex.

### 3.6. Test Design Adaptation for SARS-CoV-2 Variants

The design of the sVNT proposed in this study can be easily adapted to the change of the dominant variant of the virus; for this, it is sufficient to immobilize the corresponding mutant RBD variant on microplates. We prepared the RBD from the SARS-CoV-2 beta variant and performed the sVNT with this antigen. The RBD:ACE2-HRP binding curve retained the linear shape, but the 2.0 AU signal was obtained at the 2.9 nM ACE2-HRP concentration, reflecting a slower association rate of the ACE2 and beta-RBD. The sVNT was performed for 20 randomly chosen donor serum samples, and it was found that most of the samples are less inhibitory for the beta-RBD (Figure 5A), with only two samples showing higher ID50 toward the beta-RBD protein. We hypothesized that the ID50 drop should correlate well with the drop in the anti-RBD IgG titers. RBD ELISA test, performed for the 20 samples subset, shows some decrease in the relative IgG levels in the case of beta-RBD antigen (Figure 5B), but this drop was not the same as the ID50 drop in most of the samples (Figure 5C). The RBD ELISA testing for current and emerging virus variants cannot replace sVNT testing.

## 4. Discussion and Conclusions

It is widely assumed that neutralizing antibodies prevent re-infection with SARS-CoV-2 in recovered people or first infection in immunized people, and such antibodies may help treat patients with severe COVID-19. The humoral immune response to SARS-CoV-2 resulting from vaccination with existing types of vaccines decreases significantly over time [30,31] and is not sterilizing against the currently prevailing variants of the virus [32]. This situation requires continued large-scale testing of nAb levels in humans to identify groups and individuals resistant to the virus.

Another area of application for mass testing of the level of neutralizing antibodies is the therapy of COVID-19 with donated blood plasma. The FDA approval for the emergency use of COVID-19 convalescent plasma for the treatment of hospitalized patients with COVID-19 lists sVNT as one of the possible plasma testing methods [33]. It should be noted that most S-protein-binding antibodies, and even most RBD-binding antibodies, do not have the ability to neutralize the virus [9,34]. Total anti-RBD or anti-S antibody titer may more or less correlate with nAb levels [35] but cannot replace VNT or sVNT results.

The gold standard in the determination of the virus neutralization potential of human serum samples is the virus neutralization test, performed with the live virus and cultured human cells, expressing the specific viral receptor in the format of plaque reduction neutralization test (PRNT). This test is imprecise, costly, and labor-intensive. In the case of SARS-CoV-2, PRNT should be performed in the BSL-3 biocontainment environment, making it quite unpractical for wide use.

Replacing this testing method with other cell-based assays could reduce the cost of testing and the risk of personnel infection, but all such tests cannot be performed in a conventional clinical laboratory. Cell-based assays can only be performed in specialized laboratories and are difficult to standardize, making interlaboratory comparison almost impossible. In contrast, sVNT is performed using the same techniques and equipment as routine ELISA, so this method is suitable for widespread use. Despite its simplicity, sVNT has higher accuracy and similar specificity to more complex cell-based tests. There are reports of a higher specificity of sVNT compared to VNT in the first week after symptoms onset [36]. Other authors show the absence of cross-reactivity with SARS-CoV-1, hCoV-OC43 (β-coronavirus), and hCoV-229E (α-coronavirus) of the sVNT [37].

The main limitation of the sVNT is that it only detects neutralizing antibodies binding to the RBD area of the S-protein. Several studies have demonstrated that some neutralizing antibodies react with other areas of the S protein, lying in both the S1 and S2 subunits [38,39]. Some neutralizing antibodies bind to the RBD area of the S-protein, disrupt the viral entry into the cultured cells, but do not block the RBD:ACE2 complex formation [40]. However, despite the existence of RBD-independent antibodies that can play a role in neutralization, RBD-targeted nAbs are immunodominant [41].

Surrogate VNT designs for the SARS-CoV-2 are already described in several publications [42,43], but most researchers utilized only one set of reagents—microplates with immobilized ACE2 and the conjugated RBD in the upper layer. The reversed order of reagents layers was directly described only in two studies—the test design utilizing the biotin-conjugated RBD as the probe [18] and the label-free assay, based on the thin-film interferometry detection method [44]. In both cases, the authors claimed these tests to be suitable for routine use. The sVNT design with the ACE2-HRP conjugate was never reported to date and, therefore, the relative performance of RBD:ACE2-HRP and ACE2:RBD-HRP test designs was not compared directly.

The RBD-immobilized sVNT design has two main advantages. The first is the single-pot format—i.e., no pre-incubation of the samples with RBD outside of the test microplate. Secondly, the ease of adapting the test to different variants of SARS-CoV-2. There is no need to prepare new RBD-HRP reagents for each new virus variant. Whereas in the case of the ACE2:RBD-HRP sVNT design, new SARS-CoV-2 variants render previously manufactured RBD-HRP reagents obsolete and require new conjugate preparation and re-optimization to maintain test performance. We have experimentally shown that the RBD:ACE2-HRP test design may be successfully applied for the SARS-CoV-2 beta variant with only one test protocol change—a twofold increase in the concentration of the ACE2-HRP reagent.

Using serum samples from plasma from 73 convalescent donors with varying levels of nAbs, we have shown that the RBD:ACE2-HRP sVNT design produces results that are highly correlated (Spearman’s rho 0.855) with the VNT results. A direct comparison of the sVNT–VNT correlation strengths described in various articles [15,16,17,18,37,44] is hampered by the lack of equivalence of VNT parameters and variations in the S protein sequences of the live virus and/or pseudovirus and large variation in sample size. The strength of the correlation reported in this study is in line with the best values reported to date, indicating that the RBD:ACE2-HRP design is at least as good as the widely used ACE2:RBD-HRP test design. We expect the RBD:ACE2-HRP design to be more suitable for routine nAb testing.

The sVNT results are usually strongly correlated to the anti-RBD antibody titers [45,46], but they are not entirely equivalent. The question of the interchangeability of sVNT and RBD ELISA remains open, possibly requiring more detailed data on the interaction of antibodies with different RBD epitopes. We assume both test results (nAb levels and the anti-RBD IgG titer) could be used to construct some composite index of valuable anti-SARS-CoV-2 antibodies levels. Even a pair of these ELISA-based tests turns out to be much cheaper and easier to perform than a single test with a live or pseudotyped virus. Surrogate VNT described in this article can be used for the wide screening of vaccinated or previously infected individuals.

## Figures and Tables

**Figure 1 diagnostics-12-00393-f001:**
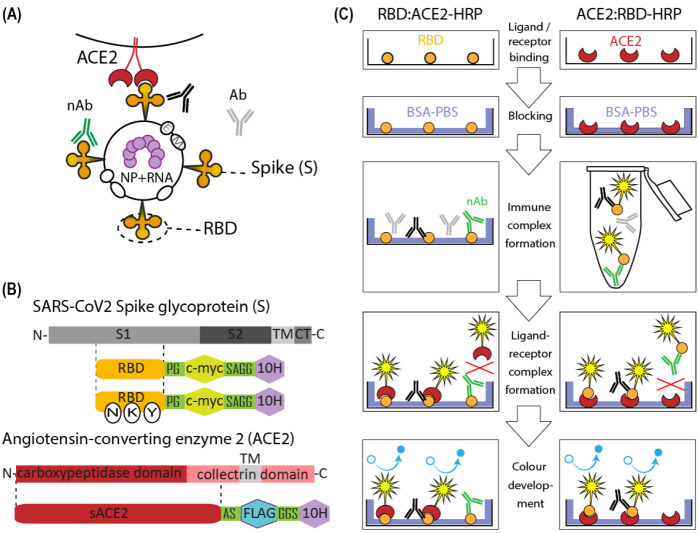
Scheme of the structure of the SARS-CoV-2, its spike protein, and cellularACE2 proteins, and the principles of the surrogate virus-neutralization test (sVNT) design variants. (**A**) Scheme of the structure of the SARS-CoV-2 and its interaction with the cell surface. Structural proteins of the SARS-CoV-2 are Envelope (E), Membrane (M), Nucleocapsid (NP), and Spike (S). NP forms complex with viral RNA shown in purple, E and M are shown as white ovals, and S trimers shown in orange. The dotted oval outlines the RBD—receptor binding domain. The RBD of the spike proteins of SARS-CoV-2 bind to the angiotensin-converting enzyme 2 (ACE2) protein (shown in red) on the cell surface. Neutralizing antibody (nAb) that blocks the RBD-ACE2 interaction is shown in green, RBD-binding non-neutralizing antibody is shown in black, non-specific antibody is shown in grey; (**B**) Schematic diagram of SARS-CoV-2 spike and ACE-2 receptor proteins structures—WT full length and truncated recombinant proteins. The spike protein (S) contains two subunits, the receptor-binding subunit S1 and the fusion subunit S2, transmembrane domain (TM), cytoplasmic tail (CT). S1 subunit includes the receptor-binding domain (RBD). The recombinant proteins used in the current work correspond to the RBD domain of the S protein with the addition of C-terminal c-myc and decahistidine tags. Amino acid linker’s sequences (PG and SAGG) are shown in single letter code. The RBD of beta-variant of SARS-CoV-2 (B.1.351, South Africa) contains three mutations K417 N; E484K; N501Y shown in white ovals. Angiotensin-converting enzyme 2 (ACE2) contains the N-terminal carboxypeptidase domain and the C-terminal collectrin domain with a transmembrane (TM) region. Recombinant ACE2 used in the current work corresponds to the soluble extracellular part of the ACE2 with C-terminal FLAG (DYKDDDDK) and decahistidine tags. Amino acid linker’s sequences (AS and GGS) are shown in single letter code; (**C**) The principles of two surrogate virus neutralization assay designs, based on the detection of RBD-ACE2 interaction that NAbs have not blocked. The left side of the figure shows an sVNT design, in which recombinant ACE2 conjugated with the horseradish peroxidase (HRP, shown as yellow star) is used as a detection reagent, and RBD is immobilized on the ELISA well surface (RBD:ACE2-HRP). The right side of the figure shows an sVNT design, in which recombinant ACE2 is immobilized and recombinant RBD conjugated with the horseradish peroxidase (HRP) is used for detection. Both sVNT design variants are realized as sandwich ELISA. Coated ELISA plates are blocked with bovine serum albumin (BSA) solution in PBS. The sera samples are added to wells (left) or pre-incubated in plastic tubes with RBD-HRP conjugates (right). Immune complexes are formed. Neutralizing antibody (nAb) that blocks the RBD-ACE2 interaction is shown in green, RBD-binding non-neutralizing antibody is shown in black, non-specific antibody is shown in grey. Then ligand–receptor complex is formed—upon addition of ACE2-HRP to the ELISA well (left) or upon addition of pre-incubated with sera RBD-HRP to immobilized ACE2 (right). The detection is performed using HRP substrate at 450 nM.

**Figure 2 diagnostics-12-00393-f002:**
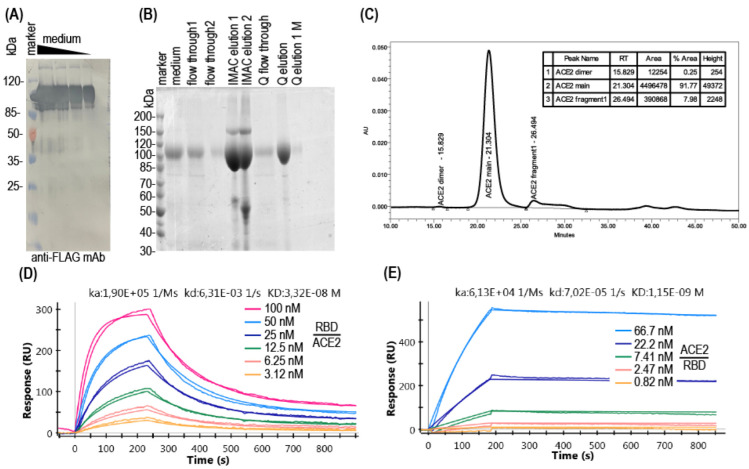
Characterization of the recombinant ACE2 reagent used for the surrogate virus-neutralization testing. (**A**) Immunoblot analysis of recombinant soluble ACE2 (sACE2) in culture medium applied equivalent of the 1000, 500, 250, 125 µL of conditioned medium/lane, reducing conditions. Anti-FLAG Ab-HRP conjugate was used; (**B**) SDS-PAGE of ACE2 during purification, reducing conditions. Flow through 1 and flow through 2—conditioned culture medium, passed through the Ni-Sepharose column one or two times. IMAC elution 1 and IMAC elution 2—fractions of first and second elution of ACE2 from the Ni-Sepharose column. Q flow through—fraction of proteins, not adsorbed on the Q Fast Flow column, Q elution—main elution peak, Q elution 1 M—fraction, obtained by the elution from Q Fast Flow column with the 1 M NaCl, 20 mM sodium phosphate solution. All samples applied as 10 µL/lane, non-reducing conditions. (**C**) Chromatography trace of the size exclusion chromatography analysis of the purified ACE2 protein. (**D**) Sensogram of the SPR analysis of the interaction of the covalently immobilized ACE2 and RBD in solution, one representative analysis lane out of three. (**E**) Sensogram of the SPR analysis of the interaction of the covalently immobilized RBD and ACE2 in solution, one representative analysis lane out of three.

**Figure 3 diagnostics-12-00393-f003:**
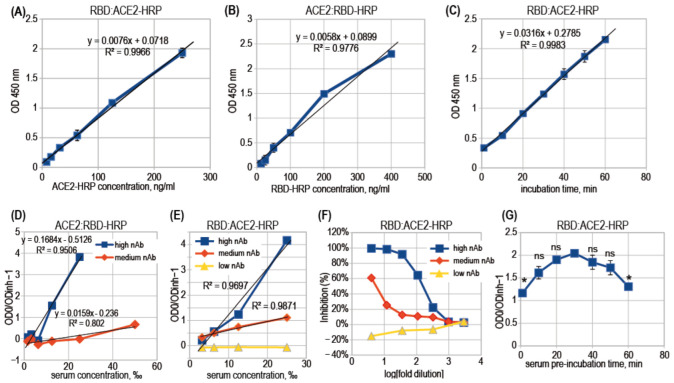
Performance comparison of two sVNT designs. The test design used is stated above for each figure’s panel. (**A**) Binding curve of the ACE2-HRP probe to the immobilized RBD, 30 min probe incubation time. (**B**) Binding curve of the RBD2-HRP probe to the immobilized ACE2, 60 min probe incubation time. (**C**) Kinetics of the complex formation between immobilized RBD and ACE2-HRP, ACE2-HRP concentration 400 ng/mL. (**D**) Inhibition curve shape for two serum samples with high and medium levels of nAb, immobilized ACE2. (**E**) Inhibition curve shape for three serum samples with high, medium, and low levels of nAb, immobilized RBD. High nAb—donor with the VNT threshold dilution 1280×; medium nAb—threshold dilution 20×, low nAb—no inhibition in the VNT. (**F**) Inhibition curve shape in logarithmic *x*-axis and inhibition percent as y. Lack of linear shape. (**G**) Variation in the observed degree of inhibition for one serum sample and various lengths of pre-incubation of test serum sample in microplate before the addition of the ACE2-HRP probe, unpaired two-sided *t*-test analysis, all data points versus 30 min pre-incubation, n = 2, * *p* < 0.05, ns—*p* > 0.05.

**Figure 4 diagnostics-12-00393-f004:**
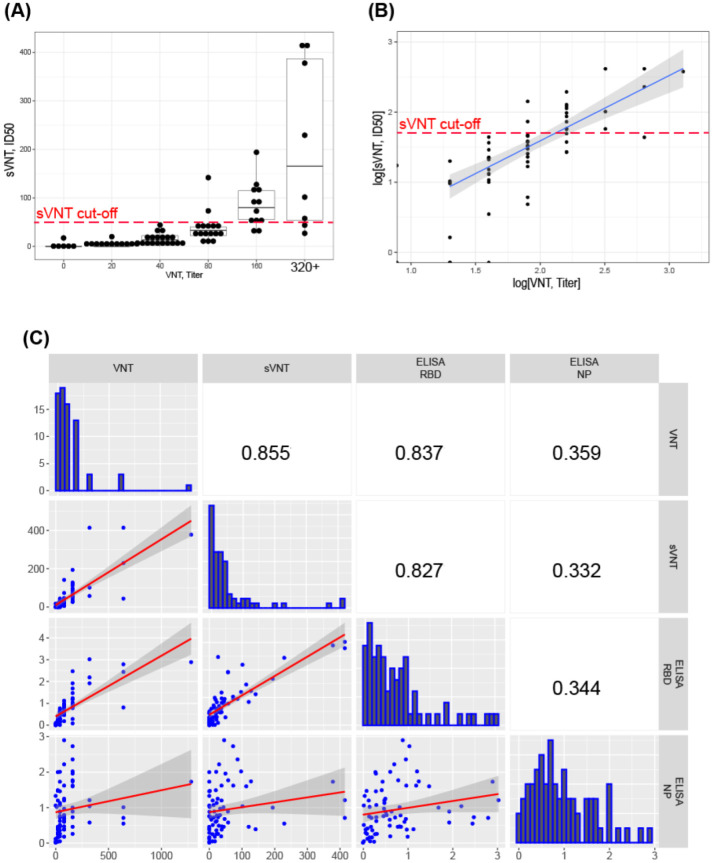
Correlation of the sVNT and VNT data, convalescent plasma samples. (**A**) Comparative analysis of individual serum samples, proposed sVNT cut-off marked with the red dashed line. (**B**) Correlation analysis of the VNT and sVNT levels of nAb, log scales. (**C**) Correlation analysis matrix for 73 COVID-19 sera with different levels of SARS-CoV-2 NAbs measured with VNT, sVNT, indirect RBD ELISA, indirect NP ELISA. Calculated characteristic values for each serum: VNT—serum dilution, corresponding to the 50% plaque reduction; sVNT—dilution, corresponding to 50% inhibition of complex formation (ID_50_); RBD ELISA and NP ELISA-OD/CO. Spearman correlation coefficients. The dark areas indicate the standard deviations of the linear regression plots. n = 2 for sVNT, RBD ELISA, NP ELISA.

**Figure 5 diagnostics-12-00393-f005:**
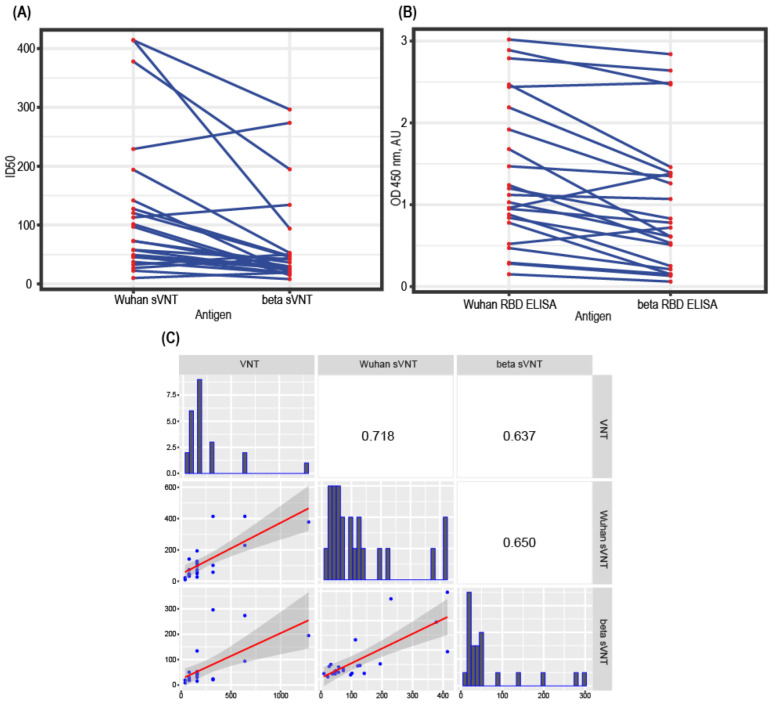
sVNT performance for SARS-CoV-2 variant beta RBD. (**A**) Change in the ID_50_ magnitudes for 20 randomly chosen serum samples. (**B**) Change in OD 450 nM in the RBD ELISA test for the same 20 serum samples. (**C**) Correlation analysis matrix for 20 COVID-19 sera—VNT with the wild-type (Wuhan) v of the SARS-CoV-2 virus, sVNT with the variant beta RBD. Calculated characteristic values for each serum: VNT—serum dilution, corresponding to the 50% plaque reduction; sVNT—dilution, corresponding to 50% inhibition of complex formation (ID_50_); RBD ELISA-OD/CO. Spearman correlation coefficients. The dark areas indicate the standard deviations of the linear regression plots. Samples were analyzed in duplicate for sVNT and RBD ELISA.

## Data Availability

The recombinant ACE-2, RBD and beta-RBD proteins described in this study is available under a standard MTA with the Federal Research Center “Fundamentals of Biotechnology” Russian Academy of Sciences. pTM, pTM-RBDv2, pTF, pTF-ACE2s are available from Addgene: #162783; #162785; #162784; 162786. Code for the correlation analysis is deposited in GitHub (https://github.com/d-kolesiko/sVNT accessed on 2 February 2022).

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
