# Peer review of "Fast and Accurate Surrogate Virus Neutralization Test Based on Antibody-Mediated Blocking of the Interaction of ACE2 and SARS-CoV-2 Spike Protein RBD"

_diagnostics, 2022, doi:10.3390/diagnostics12020393_

Round 1

Reviewer 1 Report

In this manuscript, Kolesov and colleagues proposed a new format for the surrogate virus neutralization test (sVNT). In this modification of the method, the reverse ACE2-RBD interaction is formed instead of RBD-ACE2. This simplifies the test and can also be used to evaluate sVNT of SASRS-CoV-2 mutant variants. The proposed scheme for conducting sVNT is not absolutely new. As the authors note, it was proposed earlier. However, the authors have made some additional improvements. The study is of interest and very relevant with the current pandemic. The manuscript can be recommended for publication in Diagnostics.

I do have a few minor comments about the manuscript.

A distinction should be made between antigen-antibody and receptor-ligand interaction. In Fig. 1C “Antigen binding” should be replaced with “Receptor/ligand binding”

At the first mention of pVNT, it should be noted that this is a pseudotyped VNT

In Figure 2A as well as in the caption, it is necessary to indicate the specificity of the antibodies that were used to develop Western blot. In Figure 2B, the molecular weights of the markers must be indicated. In Figure 2C, D it is necessary to indicate how the individual sensorograms differ from each other.

It is necessary to comment on why the dissociation constants in the pairs ACE2/RBD and RBD/ACE2 differed by almost an order of magnitude.

In figures 2D-F it is necessary to mark which curves correspond to three serum samples with high, medium and low levels of nAb.

Fig. 2G. The two experiments performed are insufficient for paired t-test analysis.

The fragment (lines 406 - 420), which describes the calculation of the Degree of Inhibition (DI), is written very incomprehensibly. It is advisable to either provide a reference to how the DI is calculated or to explain these calculations more clearly. Using subscripts in formula DI = (OD0 / ODinh -1) can also make it easier to understand.

I suggest the authors consider replacing Dil50% with the more familiar ID50 (inhibitory dilution).

It is necessary to explain what is OD/CO.

Author Response

We would like to thank both Reviewers for their time, effort and helpful comments. The article manuscript was updated accordingly to their requests. Reviewers’ comments and responses are as follows. In all cases we used line numbers from the original manuscript file, most of them are not the same in the revised manuscript.

Reviewer 1 comments:

  • A distinction should be made between antigen-antibody and receptor-ligand interaction. In Fig. 1C “Antigen binding” should be replaced with “Receptor/ligand binding”

Figure 1 updated – “Ligand / receptor binding” instead of “Antigen binding” in Fig 1C.

  • At the first mention of pVNT, it should be noted that this is a pseudotyped VNT

We added to the line 109 the full test name “…resulting in the pseudotyped virus neutralization test (pVNT)”. 

  • In Figure 2A as well as in the caption, it is necessary to indicate the specificity of the antibodies that were used to develop Western blot. In Figure 2B, the molecular weights of the markers must be indicated. In Figure 2C, D it is necessary to indicate how the individual sensorograms differ from each other.

Specificity of the antibodies added to the Figure 2A, below the image of the membrane, and to the Figure 2A’s caption; MW’s are added to the Figure 2B; analyte concentrations are placed on Figure 2D, 2E panels and are linked with the individual sensograms.

  • It is necessary to comment on why the dissociation constants in the pairs ACE2/RBD and RBD/ACE2 differed by almost an order of magnitude.

We added the description of the sensograms from Figure 2E to line 322: “resulting in the erroneous calculation of the dissociation constant as 1 nM due to practical absence of analyte desorption in 10 minutes of monitoring”

  • In figures 2D-F it is necessary to mark which curves correspond to three serum samples with high, medium and low levels of nAb.

We added all curve legends to Figure 2 D, E, F and added threshold dilutions in the VNT for these three samples to Figure’s legend.

  • 2G. The two experiments performed are insufficient for paired t-test analysis.

We suppose that the Reviewer mentioned Figure 3G. We performed the unpaired t-test, two-sided, and see no changes in the statistical significance – 1 min and 60 min time points are different from 30 min, rest of time points are not. We changed the legend of Figure 3 G accordingly: “unpaired two-sided t-test analysis.” Raw ELISA data were submitted to the Journal as the Supporting file; this set has the name Fig3G.xls and can be inspected for the independent statistics calculation. 

  • The fragment (lines 406 - 420), which describes the calculation of the Degree of Inhibition (DI), is written very incomprehensibly. It is advisable to either provide a reference to how the DI is calculated or to explain these calculations more clearly. Using subscripts in formula DI = (OD0/ ODinh -1) can also make it easier to understand.

We have added the subscripts to the formula. We realize that the “degree of inhibition” term is unusual, but we report that this function gives a linear dose-response relationship and is useful for data analysis. As shown in Figure 3F, the dose-response curve for the response function “inhibition %” is non-linear and requires the logistics curve analysis to determine the serum dilution, producing the 50% inhibitory effect (or half-maximum inhibitory dilution, or ID50). According to Figure 3 E, the serum concentration producing the ID50 can be calculated using simple linear regression if the “degree of inhibition” is used as the response function. In addition, as depicted in Figure A3, the calculation methodology of the sVNT response does not affect the final correlation of the sVNT and VNT data; therefore, ”degree of inhibition” can be used for comparison with other tests. We show the simple formula for the calculation of this function.

  • I suggest the authors consider replacing Dil50% with the more familiar ID50(inhibitory dilution).

We have replaced Dil50% with ID50 in the text and figures in all cases.

  • It is necessary to explain what is OD/CO.

We added the description of this function to the Methods section: “Positivity indices (OD/CO) were calculated as the ratio of the mean optical density for the test sample and the mean optical density in negative wells plus three times the standard deviation, according to the [23]”

Reviewer 2 Report

Dear authors,

Please find most of my comments within the attachment

Regards

Author Response

We would like to thank both Reviewers for their time, effort, and helpful comments. The article manuscript was updated accordingly to their requests. Reviewers’ comments and responses are as follows. In all cases we used line numbers from the original manuscript file, most of them are not the same in the revised manuscript.

Reviewer 2 comments:

  • Line 2: It is still ELISA assay, virus neutralization only when you have live virus? Better to change the title.

We have checked the usage of the “surrogate virus neutralization test” definition in the titles of published articles and found a lot of such articles:

  1. Performance Evaluation of the BZ COVID-19 Neutralizing Antibody Test for the Culture-Free and Rapid Detection of SARS-CoV-2 Neutralizing Antibodies. https://pubmed.ncbi.nlm.nih.gov/34943430/
  2. Differences in Post-mRNA Vaccination SARS-CoV-2 IgG Concentrations and Surrogate Virus Neutralization Test Response by HIV Status and Type of Vaccine: a Matched Case-Control Observational Study https://pubmed.ncbi.nlm.nih.gov/34864962/
  3. Two novel SARS-CoV-2 surrogate virus neutralization assays are suitable for assessing successful immunization with mRNA-1273 https://pubmed.ncbi.nlm.nih.gov/34563583/
  4. Evaluation of the correlation between the access SARS-CoV-2 IgM and IgG II antibody tests with the SARS-CoV-2 surrogate virus neutralization test https://pubmed.ncbi.nlm.nih.gov/34524695/
  5. Evaluation of a multi-species SARS-CoV-2 surrogate virus neutralization test https://pubmed.ncbi.nlm.nih.gov/34458548/
  6. Quantitative SARS-CoV-2 Spike Antibody Response in COVID-19 Patients Using Three Fully Automated Immunoassays and a Surrogate Virus Neutralization Test https://pubmed.ncbi.nlm.nih.gov/34441430/

In addition, the GenScript cPass™ test kit has the full name “SARS-CoV-2 Surrogate Virus Neutralization Test (sVNT) Kit”. We would like to keep the name because the term “surrogate virus neutralization test” is common and is used exactly as we have used it – as the name of the ELISA-like assay mimicking the virus neutralization test.

  • Line 22: The authors focused on ACE2 and neglect the role of TMPRSS2 however all recent studies highlight the critical role or TMPRSS2 in VOC infection and pathogenesis, please explain

We have mentioned the TMPRSS2 in the Introduction section, line 102, context “… test is based on HEK293 cells, overexpressing … the membrane-bound serine protease TMPRSS2 [13], which cleaves the S-protein and primes its conversion to the membrane-protruding conformation”. The site of S-protein cleavage by the TMPRSS2 lies outside the RBD domain, so the sVNT test is insensitive to TMPRSS2 cleavage. Generally, the prevalence of one from the two major SARS-CoV-2 internalization routes is complex, and some cell-based models show that endocytosis and subsequent cathepsin cleavage of the S-protein dominate over the direct membrane fusion mediated by the TMPRSS2. Other models and new VOCs demonstrate the prevalence of the TMPRSS2-mediated entry. Both ways are ACE2-dependent; this article reports on the preferable assay design for measuring the inhibition of the RBD:ACE2 complex formation and not on the internalization blocking. As we (and many other researchers) have shown, inhibition of the RBD:ACE2 binding in most cases correlates with the real virus neutralization in cell culture. We changed the TMPRSS2 description at line 100 as follows “For the SARS-CoV-2 virus, PRNT is usually based on HEK293 cells overexpressing the cellular receptor ACE2 [12], which binds to the viral S protein, and in some cases additionally overexpressing the TMPRSS2 protease [13]. This membrane-bound serine protease cleaves the S-protein and initiates its conversion to a membrane-protruding conformation.”

  • Line 26: How you justify about good or not good? How about the statistical analysis?

We changed line 26 according to the request – “its results showed a very strong correlation with VNT (Spearman’s Rho 0.83)”. We defined Spearman’s Rho meanings in line 292; these meanings are very common. Statistical analysis is described in Materials and Methods, lines 289-295. Code for the correlation analysis performed is deposited in GitHub (https://github.com/d-kolesiko/sVNT).

  • Line 26: 3 amino acids mutation compared to which variant?

Compared to the Wuhan reference variant Wuhan-Hu-1. We changed the Abstract, line 21, and introduced the name of the SARS-CoV-2 variant used: “intact RBD from the Wuhan-Hu-1 virus variant”.

  • Line 29: There is no clear conclusion

We added to line 29 one more phrase: “The sVNT assay design with the ACE2-HRP is preferable over the assay with the RBD-HRP reagent and is suitable for mass screening of neutralizing antibodies titers.”

  • Line 115: Is that only restricted to certain variant or majority of VOCs?

We rewrote the phrase in line 115 to make it more transparent: “The surrogate VNT (sVNT) measures the blocking of the ligand-receptor complex formation upon the addition of the test serum samples.” Like all other methods, the sVNT will produce different numbers with various RBD variants, and we illustrated this in Figure 5. We are sure that all VNT tests show different results with VOCs; this topic is at the edge of current SARS-CoV-2 immunity research, and the magnitude of nAb titers change for delta and omicron variants is widely debated, but the conclusions of published studies are usually based on rather small and inclined datasets. We aimed to introduce the methodology of a really simple and cheap testing, which, in our opinion, could help to produce much larger datasets for COVID-19 epidemiology analysis and predictions.

  • Line 116-120: This is methodology not introduction

We consider this short sentence to be an important transition from a review of neutralizing antibody tests to a detailed description of our methodology. It also contains critical notes about the speed of testing and the required level of laboratory biosecurity. We rephrased this text as follows: “For the SARS-CoV-2 virus, PRNT is usually based on HEK293 cells overexpressing the cellular receptor ACE2 [12], which binds to the viral S protein, and in some cases additionally overexpressing the TMPRSS2 protease [13]. This membrane-bound serine protease cleaves the S-protein and initiates its conversion to a membrane-protruding conformation.”

  • Line 131: Is it concluded as competitive ELISA here?

We agree that sVNT is sometimes similar to the competitive ELISA. However, the underlying math is different – we do not try to determine the dissociation constant by adding the receptor to the ligand solution and establishing the equilibrium in the microplate wells. We block the ligand on the microplate by the antibodies and measure the residual level of the unblocked ligand by the slowly binding receptor-HRP conjugate. This description, of course, is very simplified. We made no changes to the article text in this case.

  • Line 140: It is highly recommended to compare this assay result to SNT assay using Beta SARS CoV2

Unfortunately, we cannot answer this question properly – we have no live beta variant SARS-CoV-2 virus on hand, and we suppose this variant to be practically extinct in nature by now. It was the actual VOC at the time of experiments were performed and was included in the article as the illustration of various Ab titer changes for RBD ELISA and sVNT testing. We will continue sVNT testing with the omicron's variant RBD, now in preparation, and plan to report the results in a separate article.

  • Line 150: Better to put this table as supplementary

We moved the Table 1 to the Appendix section, Table S1.

  • Line 150-170: This section need to be written again, it is confused and not organized

We rewrote this section completely.

  • Line 177: Can you add a reference here?

We added the reference to the old Chasin and Urlaub article here. doi:10.1073/pnas.77.7.4216

  • Line 206: Better to describe the followed protocol, will be helpful for the readers, also add a reference

We added a more detailed description of the medium ultrafiltration procedure; the rest of the analysis is quite common – just the SDS-PAGE and Western blotting. We referred to our previous article, where this analysis is described in detail.

  • Line 257: You add t in title but nothing mentioned below about SNT

We suggest that the question was about the VNT with live SARS-CoV-2 virus; this test is equivalent to the Serum Neutralization Test. We added a more detailed description of the VNT in line 257: “Live hCoV-19/Russia/Moscow_PMVL-1/2020 virus variant was employed; test was performed as the microneutralization assay with the Vero E6 cells. In total, 73 samples were used for current study. This virus variant contains one amino acid substitution in the S-protein RBD relative to the Wuhan variant – 614D>G.”

  • Line 258: Please mention the ethical approval of sera samples collection

We added the ethical approval data to line 258.

  • Line 268: This is the first time you mention NP?

No, we added the description of the NP earlier, line 227, and set the abbreviation “NP” for the first time there.

  • Line 326: Figure2A and 2B resolution is too low, and need to be improved

We apologize for the ugly view of the 2A and 2B; this problem is due to extensive downsampling of the embedded bitmap images by the Journal’s manuscript processing engine. Perhaps it is possible to download Figures files as they were uploaded, recheck the bitmaps, or extract the original bitmap images from the raw data zip file attached to the manuscript. We are sure that we have uploaded both manuscript-embedded and standalone pictures with sufficient resolution.

  • Line 427: Sera samples still a major issue?

Yes, of course. We have a limited capacity for the live virus VNT and strict limitations for the sera use outside of the routine testing. It took us a while to receive ethical approval for these 73 donor plasma samples.

  • Line 466: All figures are not discussed properly within the result section? Please try to discuss more about results

We have re-checked the references to the Figure' panels in the text and added the reference to Figure A 1B (appendix). We agree that Figure 1A is referenced only in the Introduction section; it is not referenced from the Results section. This panel contains the SARS-CoV-2 picture with protein names, and we believe it is appropriate to refer to this panel only in the "Introduction" section; it is needed for a visual representation of the protein names used in the text of the article. The rest of the Figure' panels are referenced and discussed in the Results section.

  • Line 475: Authors did not discuss their results properly compared to previous studies and to show the novelty of their study? Discussion need to be written again.

We have found only two articles describing the equivalents of the RBD:ACE2-HRP test design, no articles for the exact RBD:ACE2-HRP design, and no articles with a direct comparison of both designs. We think that by this article, we will close a significant gap in the routine nAb testing methodology; we mentioned in the Discussion that we report on direct comparison of two sVNT designs for the first time and have found them to be strikingly unequal. We made all these statements in the Discussion section.

Unfortunately, a more detailed comparison of our results and previously published sVNT-based results is impossible. As we have mentioned in the Abstract and Introduction sections (lines 135-137), published research articles report vastly different correlation levels of sVNT and VNT (or pVNT), possible reasons for these differences were already explained in the Introduction section (lines 139-149). An example of such discrepancies can be found for the Genscript Cpass test in the articles [15] and [17] – correlation levels 0.84 and 0.49 for nearly the same reference VNT tests and the same sVNT test. We would like to describe another major reason for such discrepancies – suboptimal sVNT assay design. We discuss in detail the relative performance of the usual and "inverted" sVNT; this topic is, in our opinion, the most relevant one. We rewrote the Discussion section completely, phrase by phrase, and suppose it to be clearer.

Round 2

Reviewer 2 Report

Thanks for considering my comments. The manuscript still require moderate English editing. Please note that Wuhan is not variant